# Genomic Insights into Cultivated Mexican *Vanilla planifolia* Reveal High Levels of Heterozygosity Stemming from Hybridization

**DOI:** 10.3390/plants11162090

**Published:** 2022-08-11

**Authors:** Paige Ellestad, Miguel Angel Pérez-Farrera, Sven Buerki

**Affiliations:** 1Department of Biological Sciences, Boise State University, 1910 University Drive, Boise, ID 83725, USA; 2Herbario Eizi Matuda, Laboratory of Evolutionary Ecology, Institute of Biological Sciences, Universidad de Ciencias y Artes deChiapas, Libramiento Norte Poniente 1151, Col. Lajas Maciel, Tuxtla Gutiérrez 29039, Mexico

**Keywords:** conservation, domestication, heterozygosity, hybridization, k-mer, landraces, Mexico, Orchidaceae, vanilla, whole genome resequencing

## Abstract

Although vanilla is one of the most valuable spices, there is a lack of understanding of the genomic variability of the main vanilla producing species, *Vanilla planifolia,* within its cultivated origin, Mexico. High genomic heterozygosity levels within the globally cultivated ‘Daphna’ genome have raised questions on the possibility of a hybrid origin and analogous genomic signatures of vanilla cultivated within its origin. This study investigated these questions by assessing whether the genomic structure of Mexican *V. planifolia* reflected domestication events. Whole genome re-sequencing was used to compare genome complexity between 15 cultivated accessions from different regions and gene pools. Results showed high levels of heterozygosity, ranging from 2.48% to 2.85%, in all but one accession, which exhibited a low level (0.403%). Chromosome-level comparative analyses revealed genomic variability among samples, but no signals of chromosome rearrangements. These findings support the hypotheses that cultivated vanilla resulted from hybridization and that multiple domestication events have shaped cultivated vanilla leading to the formation of landraces. High cultural diversity within this region further supports the occurrence of multiple domestication processes. These results may help to improve breeding and conservation efforts aiming to preserve the genetic diversity of this beloved spice threatened by climate change.

## 1. Introduction

*Vanilla planifolia* Andrews is a tropical vine of the family Orchidaceae, which produces vanilla, one of the most widely known and valuable spices worldwide [1]. With a cultivated origin in Mexico, it has been introduced across the globe to be cultivated for use in the culinary, cosmetic, and medicinal industries [1,2]. By country, vanilla production is currently led by Madagascar, followed by Indonesia then Mexico [3]. *Vanilla planifolia* is self-compatible, but incapable of self-fertilization without natively co-occurring pollinators [4]. Outside of its native range, a labor-intensive technique is required to manually pollinate the flower [5]. Inhibiting natural genetic recombination, manual self-pollination and clonal vegetative propagation practices have resulted in low genetic diversity within the cultivated species, overall hindering its ability to cope with changing environmental conditions [2,4,6,7,8]. On top of increasing drought conditions [9] and fungal outbreaks associated with climate change [10], the rapid loss of wild populations due to land-use change, habitat fragmentation, and illegal harvesting poses an immediate and irreversible threat to the preservation of genetic variation within this crop [8,11]. Genetic resources within *V. planifolia’s* cultivated center of origin may offer a novel gene pool to increase the genetic diversity within the species and ensure crop sustainability under future climate scenarios. Analyzing the genomic structure of regionally cultivated vanilla in Mexico will offer important insight into this crop’s genetic resources and a better understanding of the processes leading to its domestication.

Ancient and contemporary cultural groups have shaped vanilla in its center of cultivation for centuries. Historical records indicate that vanilla was used as a flavoring and medicinal beverage by multiple cultures in Mesoamerica, including the Totonacs, the Mayans and the Aztecs [1,12]. After the Spanish conquest of the Aztecs in 1520 AD, it was transported to Europe, but was not cultivated outside of its native range until 1832, when Edmond Albius, from Reunion Island, developed a technique for manually pollinating the flowers [1,13]. This human-mediated dispersal has led many researchers to believe that globally cultivated vanilla (i.e., cultivated outside of the species’ native range [14]) comes from a single origin in Mexico, specifically in the Papantla region, and this hypothesis has been supported by genetic data [2,4,15]. Within Mexico, however, high levels of genetic variability have led to the hypothesis of multiple origins shaping regionally cultivated vanilla [2,16,17,18,19], although these limited results have not been able to fully disentangle the native crop’s evolutionary history. Challenges, rooting from unclear species boundaries, intra-specific phenotypic variability, and congeneric hybridization, have hindered an accurate understanding of the processes that have shaped the genetic resources of vanilla in its origin. Additionally, the cultivation of multiple *Vanilla* species in Mexico [19], which exhibit similar vegetative morphological characteristics, muddles inferences on the genetic resources of the main vanilla producing species, *V. planifolia.*

Recent advances in genomic sequencing technology and the publication of a reference genome [6,20] have helped to elucidate vanilla’s genetic resources and uncover greater levels of genetic variation than previously expected [6,7,20,21], providing more insights into the domestication processes that have affected vanilla cultivated both in its native and its global range. Various methods to infer genetic variation have exposed high levels of variability within *V. planifolia.* Single nucleotide polymorphism (SNP) analyses have revealed variation, clustering vanilla accessions into three main groups (types 1–3), with accessions cultivated in Mexico clustering into only two [7]. Furthermore, within cultivated *V. planifolia* in Mexico, haplotype variation, inferred using ITS, was revealed, uncovering ten different haplotypes [19]. At the population level, a clear demarcation in observed heterozygosity (Ho) was found between cultivated and wild *V. planifolia*, where cultivated vanilla exhibited substantially higher levels [21].

For examining an organism’s genetic diversity and evolutionary history, genome-wide patterns of heterozygosity offer a valuable metric. Using GenomeScope, a recently developed software designed to assess the relative abundance of homozygous and heterozygous sequences within k-mer frequency distributions [22,23], recent studies have reported genome-wide heterozygosity levels of globally cultivated *V. planifolia* to be 2.32% [6] and 2.47% [20], therefore suggesting this species to be highly outbred. These high levels found within *V. planifolia* cultivated outside of its native range raise the questions of what evolutionary processes contributed to this genomic structure and whether vanilla cultivated in its origin exhibits the same genomic signals. It has been hypothesized that these high levels of heterozygosity within cultivated vanilla were attributed to the accumulation of somatic point mutations brought about by clonal propagation [21], as shown in *Manihot esculenta* Crantz [24]. The extent of these levels, however, points to the contribution of additional, more effecting, evolutionary processes, such as hybridization and/or polyploidization.

Hybridization has previously been suspected as a contributing agent to phylogenetic incongruences between nuclear and plastid signals [19] and chromosomal abnormalities [25,26] within cultivated vanilla, and may additionally offer an explanation for these high levels of heterozygosity. Hybridization has been shown to occur between *V. planifolia* and six species: *V. pompona* Schiede [20], *V. aphylla* Blume [21], *V. odorata* C. Presl, *V. ×tahitensis* J.W. Moore [18], *V. phaeantha* Rchb.f. [6], and *V. palmarum* Salzm. ex Lindl [22]. Owing to the ancient and contemporary cultivation histories in Mexico, synthetic hybridization events between local congeners is a likely possibility. On the other hand, natural polyploidization has also been shown to occur within cultivated *V. planifolia* [27] and could explain unexpectedly high levels of genome-wide heterozygosity within some individuals, although it is unlikely that these phenomena would occur in widespread cultivation.

Within this study, we aimed to explore the evolutionary mechanisms underpinning the high levels of genome-wide heterozygosity in vanilla and shed light onto the evolutionary processes that have affected this crop in its cultivated center of origin, Mexico. Due to the phenotypic and genetic variation observed in Mexico and the long histories of regional cultivation by different ethnic groups, we hypothesized that the gene pool of cultivated vanilla in Mexico has been influenced by multiple domestication events. On top of that, due to the extent of genome-wide heterozygosity levels found within globally cultivated vanilla, we hypothesized that cultivated vanilla stems from a hybrid origin. To assess if the genome structure of regionally cultivated haplotypes reflects domestication processes, we compared the genome structure of regionally cultivated vanilla against the available reference ‘Daphna’ genome [20], evaluating genome-wide heterozygosity, ploidy, synteny, and SNP relatedness. To obtain a reference scale of genome-wide heterozygosity levels in plants, we first conducted a literature review to extract all genome-wide heterozygosity values inferred using the software GenomeScope and GenomeScope 2.0 [22,23]. Our sampling consisted of 15 plants cultivated around the main vanilla producing regions of Veracruz and Oaxaca, Mexico and encompassed the breadth of haplotypic and phenotypic diversity as inferred by Ellestad et al. [19]. Genomic insights into cultivated *V. planifolia* in its origin will help shed new light on the domestication processes and genetic resources of this beloved spice threatened by climate change.

## 2. Results

The data and reproducible workflow (the code, including citations, and versions of all packages) associated with this study are available on GitHub [28], and a companion GitHub Pages website [29] was developed to fully explain our analyses.

### 2.1. Review of Plant Levels of Genomic Heterozygosity Inferred Using GenomeScope

The query for studies that have used GenomeScope to infer genomic heterozygosity resulted in 455 publications deposited on PubMed, of these 142 pertained to plants (Appendix A). For all plants assessed, the average level of genomic heterozygosity was found to be 1.59% (min 0.04%, max 12.02%; Figure 1A). For just diploid plants, the average was found to be 1.10% (min 0.07%, max 4.48%). Over half of the plants assessed in these studies were cultivated for human use (Figure 1B) and Orchidaceae was only represented by three other species (Figure 1A).Therefore, it should be noted that these values may offer a skewed scale of heterozygosity levels since genomic research on cultivated plant species often employs inbred and/or solely diploid accessions for genomic sequencing to effectively perform genomic tasks, such as read mapping and alignment. Nonetheless, the previously reported genomic heterozygosity levels (2.32% and 2.47%) for diploid *V. planifolia,* a predominantly vegetatively propagated crop, were comparatively high [6,20] (Figure 1A).

### 2.2. Sampling, DNA Extraction, and Whole Genome Re-Sequencing

Fifteen samples were collected from eight municipalities within the main cultivation regions in Mexico (Figure 2). Samples included 10 haplotypes inferred from ITS haplotype analyses [19]. Thirteen samples exhibited the most common ‘Mansa’ phenotype and three samples exhibited ‘Variegata’ phenotype, with yellow and green striped leaves, as described by Soto Arenas and Dressler [8]. Whole genome re-sequencing of quality extracted genomic DNA (see Methods for DNA concentration threshold) resulted in an average of 195 million reads (paired-end), yielding an average of 5.861 Gb per sample. Re-sequenced samples were found to have an average sequencing coverage of 80X (Table 1) to the reference genome *V. planifolia* ‘Daphna’, which had a genome length of 736,752,966 bp [20]. After trimming, samples had an average size of 13.4 Gb per paired-end read (Table 1).

### 2.3. Genomic Heterozygosity, Ploidy, and Complexity

Output from GenomeScope 2.0 analyses conducted on cleaned trimmed reads revealed similar genome-wide heterozygosity levels between the reference ‘Daphna’ genome and most Mexican *V. planifolia* samples, but a strong divergence was revealed with one Mexican sample, MEX67 (Figure 3 and Appendix A). Within 14 Mexican *V. planifolia* samples, heterozygosity levels were high, ranging from 2.48% to 2.85%. Contrastingly, MEX67 exhibited a much lower heterozygosity level of 0.403% (Table 2). Haploid genome size estimations all ranged between 513 Mbp and 613 Mbp. Heterozygous k-mer pair coverage distributions from Smudgeplot revealed signals of diploidy in all samples with an average of 97.3% of k-mer pairs in an AB ratio (Figure 4). Other ratios AABB, AAB, and AAABB were also found, but only in small percentages (<5%; Figure 4 and Appendix A).

### 2.4. Genome Reconstructions to Infer Structural Variation and Synteny

Genomic alignments from MiniMap2 [30] revealed that all reconstructed genomes exhibited full coverage on the ‘Daphna’ reference genome (Figure 5) and, in congruence with results from the dotplot analyses (Appendix A), suggested no chromosomal rearrangements among the accessions. Overall, genomic comparisons to the reference genome revealed a variation in structural similarities among Mexican samples, with MEX67 exhibiting the most similarities (Figure 6 and Appendix A). Genomic synteny to the reference genome, visualized using ‘dotPlotly’ (Appendix A), showed mean percentages of identity between 99.0 and 99.6% on all chromosomes of MEX67, while the rest of the samples showed much lower percentages of identities (98.4–99.0%). Among all samples, chromosome two (CM028151.1) matched the least to the reference genome (Figure 6 and Appendix A). Variation among samples was best visualized by the heat map of relative percentage identities by chromosome and further exemplified the extent of differences between most samples and the reference, especially on chromosome two (CM028151.1; Figure 6). Samples clustered into three main groups: the first consisting of MEX67; the second consisting of MEX65, MEX51, MEX79, and MEX41; and the third consisting of the remaining samples. Samples did not cluster by geography. Relative to other samples, MEX67 showed remarkable similarities to the reference genome.

### 2.5. SNP Calling and Clustering Analyses

A total of 7,468,839 high-quality biallelic single nucleotide polymorphisms (SNPs) were detected among all samples. Pruning for linkage disequilibrium (LD) at thresholds 0.2 and 0.8 reduced the number of filtered SNPs to 9297 and 419,885, respectively. Independently of LD thresholds, results from the principal components analysis (PCA) remained similar (Figure 7A and Appendix A). With a LD threshold set to 0.2, the top two eigen vectors explained 10.39 and 8.16% of variance (Figure 7A). Within this more conservative PCA (Figure 7A), most samples clustered together exhibiting slightly more variability in eigen vector 2 values than in eigen vector 1 values. Two samples, MEX31 and MEX67, did not cluster with the rest, nor each other. MEX31, exhibited distinctively low values along eigen vector 2 and MEX67 exhibited distinctively high values along eigen vector 1. Within the PCA set with a LD threshold of 0.8, MEX31 clustered with the other samples along eigen vector 2, but MEX67 remained distantly separated. At both thresholds, SNPs were scattered across all chromosomes, but were most numerous on chromosome 2 (Table 3, Appendix A). When SNP density was mapped onto the 14 chromosomes (using a 500 kb sliding window), non-randomly distributed SNP hotspots were revealed (Figure 7B). The most prominent hotspot occurred along a terminal region of chromosome 2; other notable hotspots occurred on terminal regions of chromosomes 9 and 14 (Figure 7B).

## 3. Discussion

### 3.1. High Genome-Wide Heterozygosity Stemming from Hybridization

Compared to other plants (Figure 1A), the reference ‘Daphna’ genome and all Mexican *V. planifolia* samples exhibited high genome-wide heterozygosity levels, except one, MEX67, which exhibited a notably low level (0.403%), (Figure 3). Similar levels have been reported in diploid *Artemisia tridentata* Nutt. (Asteraceae) and, based on the abundance of AAB (26%) and AABB (14%) k-mer ratios along with the dominant AB (49%) k-mer pairs [31], were attributed to past polyploidization followed by diploidization events within the evolution of the species. For *V. planifolia*, however, results from Smudgeplot confirmed the diploid status of all accessions by uncovering almost exclusive AB k-mer pairs (average 97.3%; Figure 4), indicating the prominence of two sub-genomes, and therefore ruling out these latter evolutionary events as sources of the observed high heterozygosity. Together, these findings support the hypothesis of a past hybridization event within cultivated *V. planifolia*. Concordantly, a hybrid origin has been previously proposed based on the evidence of chromosomal structure [25] and contrasting phylogenetic signals between chloroplast and nuclear DNA sequences [14]. Previous hypotheses have attributed these levels to the accumulation of somatic point mutations [21], however, the extent of this heterozygosity as compared to other plants (Figure 1A) suggests the occurrence of a more extreme evolutionary event, like hybridization between *V. planifolia* and a congener or two genetically differentiated *V. planifolia*. Similar levels of heterozygosity (2.27%) were observed in *Litchi chinensis* Sonn. and were attributed to the hybridization of two distinct haplotypes [32]. In addition to distinct haplotypes of *V. planifolia,* candidate parental species may include other less cultivated species like *V. pompona* or *V. odorata*. One sample, MEX67, did not exhibit the genomic signal of hybridization as the others did (Figure 3B), and therefore may represent the most in-bred form of *V. planifolia.* Further research is needed to understand the implications that these genomic structures have on fitness or other desirable traits.

### 3.2. Comparative Chromosomal Analyses Suggest Multiple Domestication Events in Mexico

Varying chromosomal structure (Figure 6), as compared to the reference ‘Daphna’ genome, and clustering patterns based on SNP relatedness (Figure 7A) suggest that multiple evolutionary pathways have shaped the genomes of cultivated Mexican *V. planifolia* leading to their similar, but variable, levels of heterozygosity. These results indicate that multiple haplotypes exist within the AB sub-genomes identified through Smudgeplot analyses (Figure 4). The accessions within this study are most likely not clones, but the result of several domestication events in Mexico. One sample, MEX67, exhibited notable differences to all other samples as shown by a substantially lower genomic heterozygosity levels (Figure 3), high degree of similarity to the reference chromosome (Figure 5 and Figure 6), and distant positioning in the PCA (Figure 7A). Largely congruent chromosomal structuring indicates similarity to the reference genome, but the conflicting heterozygosity levels contradict this similarity. Therefore, it is probable that MEX67 matches to the one haplotype that is referenced in the *V. planifolia* ‘Daphna’ genome, but not the other haplotype, which is not referenced. This sample, which was cultivated from a wild source in the Chinantla region of Oaxaca, may represent the true *V. planifolia*, from a natural non-hybrid origin.

Not considering MEX67, the two groups of samples on the chromosomal similarity heat map (Figure 6) and the clustering of most samples, except MEX31, in the PCA (Figure 7A) support the hypothesis that multiple evolutionary or domestication processes have affected vanilla cultivated in this region. Grouping of samples within the heat map did not reflect geography. The distribution of these groups throughout the entire sampling region and the additional, but less extreme, chromosomal variation within groups shows that these groups have been dispersed by humans throughout the entire sampling region and that additional domestication processes like introgression and/or the accumulation of somatic point mutations may have contributed to their genomic makeup. Although grouping within the heat map is not completely mirrored by the PCA, these results show that at least two main domestication events of hybridization have occurred within vanilla cultivated in Mexico. Considering the long histories of cultivation by groups such as the Aztecs, Mayans, and Totonacs, the findings prove reasonable in that ancient cultures might have separately influenced the genomic make-up of regionally cultivated species through the passing down of cultivation knowledge and plant material.

### 3.3. Conservation of Mexican Vanilla Landraces and Implications for Production

Diverse landraces within a crop’s native distribution provide an important source of genetic diversity to potentially increase its capacity to cope with environmental change [33]. The genomic signals of multiple origins of cultivated vanilla within Mexico support the hypothesis of landrace cultivation, which was previously suggested based on ITS haplotype analyses [19]. Additionally, results from this study suggest that most cultivated vanilla comes from a hybrid origin between either two genetically differentiated *V. planifolia,* or between *V. planifolia* and another species. Other species found in cultivation such as, *V. ×tahitensis* [34], *V. pompona*, and *V. insignis* [19], may offer parental candidates for cultivated vanilla. Given that only *V. planifolia* and *V. ×tahitensis* are recognized for commercial vanilla production [35,36], the reconsideration of species’ requirements to include congeneric species may offer novel alternative sources for vanilla production and catalyze more inclusive conservation strategies for *Vanilla*. The prioritization of agricultural diversity and the conservation of landraces within this biologically, culturally, and economically important region, will not only benefit global vanilla production and sustainability, but will also benefit the livelihoods of farmers and may help to encourage the protection of cultural diversity in Mexico.

## 4. Materials and Methods

A more comprehensive, reproducible workflow (including code, citations and package version) of methods within this study are available on GitHub [28]. Additionally, a companion GitHub Pages website [29] was developed to fully explain our analyses.

### 4.1. Review of Plant Levels of Genomic Heterozygosity Inferred Using GenomeScope

To obtain a reference of plant genomic heterozygosity levels inferred using GenomeScope [22,23], a literature review was conducted using the R package ‘easyPubMed’ [37] and ‘rentrez’ [38] querying all studies that have used this software (using the two PubMed accession numbers associated to publications related to GenomeScope) since 29 March 2022 and are deposited on PubMed. From each study, the following attributes were manually recorded by inspecting publications: species, ploidy levels, genomic heterozygosity, and estimated genome size. Additionally, a list of possible human uses for each species was obtained using categories provided in the World Checklist of Useful Plant Species, compiled by the Royal Botanic Gardens, Kew (UK) [39].

### 4.2. Sampling, DNA Extraction, and Whole Genome Resequencing

Samples were collected from 15 *Vanilla planifolia* plants within the origin of vanilla cultivation in Mexico in October of 2019 from the northernmost region around Papantla, Veracruz to the southernmost region around Chinantla, Oaxaca (Figure 2). Samples included the breadth of genetic, phenotypic, and climatic variation as inferred from ITS haplotype analyses in Ellestad et al. [19]. Voucher specimens for these individuals were deposited at Herbario Eizi Matuda (HEM) and are represented as a living collection maintained in Berriozábal, Chiapas, Mexico. From each individual, vegetative cuttings were taken, and one gram of leaf material was dried in silica gel for genomic analyses. Additionally, the publicly available phased *V. planifolia* ‘Daphna’ genome (BioProject ID: PRJNA633886), downloaded from the National Center for Biotechnology (NCBI) website, was used as a reference in this study.

Genomic DNA was extracted from all lyophilized leaf samples using the Qiagen DNeasy Plant Mini Kit (Hilden, Germany) following the manufacturer protocol. DNA yield was quantified using a Qubit fluorometer (Thermo Fisher Scientific, Inchinnan, UK). Extracted genomic DNA with concentrations greater than 20 ng/µL was sent to GENEWIZ, Inc. (South Plainfield, NJ, USA) for library preparation and sequencing of 150 bp paired-end reads using an Illumina HiSeq platform aiming for a sequencing depth of 50X to allow for sufficient coverage on the reference ‘Daphna’ genome (736,752,966 bp) [20]. Raw sequences were checked for quality using FASTQC (https://www.bioinformatics.babraham.ac.uk/projects/fastqc/ accessed on 5 May 2020) and all reads were cleaned and trimmed using Trimmomatic [40] with minimum length (MINLEN) reads set to 100 bp and a Phred score of 33.

### 4.3. Genomic Heterozygosity, Ploidy, and Complexity

Genomic sequences were characterized (size, heterozygosity, repetitiveness) by k-mer frequency analyses (k = 21) using Jellyfish [41] and GenomeScope 2.0 [22]. Using k-mer (k = 21) histograms obtained by KMC3 [42], heterozygous k-mer pairs were analyzed through Smudgeplot [22] to estimate ploidy levels and infer genomic complexity. Lower (L) and upper (U) end cut-off values, below and above which all k-mers were discarded as errors, were set using k-mer coverage output from Genomescope 2.0, as recommended in the Smudgeplot documentation (https://github.com/KamilSJaron/smudgeplot accessed on 2 April 2022) using k-mer coverage (kcov) values inferred from GenomeScope, where L = (kcov/2) − 5.

### 4.4. Genome Reconstructions to Infer Structural Variation and Synteny

Chromosome-scale genomes were reconstructed by mapping cleaned, trimmed reads to the reference *V. planifolia* ‘Daphna’ genome using Bowtie 2 [43]. Variants were called, filtered, and normalized and consensus genome sequences were created using SAMtools and BCFtools 1.15.1 [44]. Chromosome-level genome sequences were compared against the reference genome using Minimap2 [30] to assess similarity. In R [45], chromosomal coverage was evaluated using the ‘pafr’ package [46] and chromosomal rearrangements and synteny were assessed using both ‘pafr’ and ‘dotPlotly’ packages (https://github.com/tpoorten/dotPlotly accessed on 1 June 2022). For visualization of genomic variability among samples, a heat map was produced in R using ‘gplots’ [47] to show the percentage of identities between each sample and the reference genome at a chromosome level.

### 4.5. SNP Calling and Clustering Analysis

Reconstructed genomes were analyzed using BCFtools [44] to call and filter variants with Phred quality scores greater than 20. Using the R package ‘SNPRelate V1.6.4’ [48], indexed calls were further filtered to include only biallelic SNPs in linkage equilibrium with each other. Since the population processes affecting this species as a cultivated plant is unclear, linkage disequilibrium thresholds were set to the wide-ranging values 0.2 and 0.8. Also using ‘SNPRelate’ [48], principal components analyses were conducted with both linkage disequilibrium thresholds to observe and minimize the effect of SNP clusters. Results were plotted using the top two eigenvectors explaining the largest percent of variance among the data. Additionally using both linkage disequilibrium thresholds, SNP density along a 500 Kb sliding window was mapped onto the 14 chromosomes. To observe the chromosome level distribution of SNP hotspots, results were plotted using the R packaFges ‘seqinr’ [49] and ‘RCircos’ [50].

## Figures and Tables

**Figure 1 plants-11-02090-f001:**
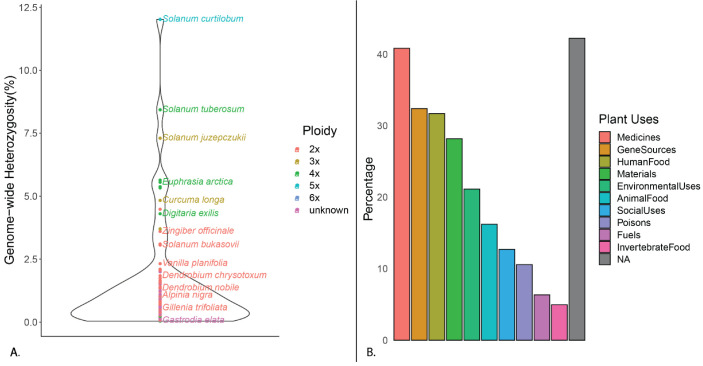
Summarized data of plant attributes extracted through a literature review of studies using GenomeScope and Smudgeplot (**A**) Violin plot of genome-wide heterozygosity levels. (**B**) Bar plot of human uses for each species studied.

**Figure 2 plants-11-02090-f002:**
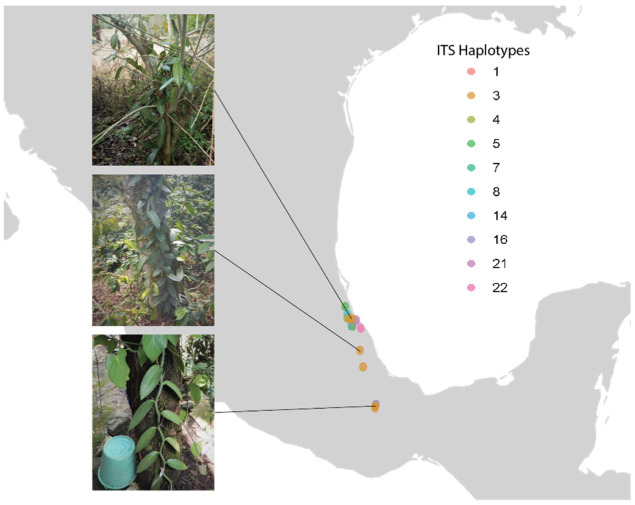
Map of sampling locations in Mexico. Point colors show haplotype IDs of samples as inferred by Ellestad et al. [19]. From top to bottom, photos show samples MEX41, MEX59, and MEX67.

**Figure 3 plants-11-02090-f003:**
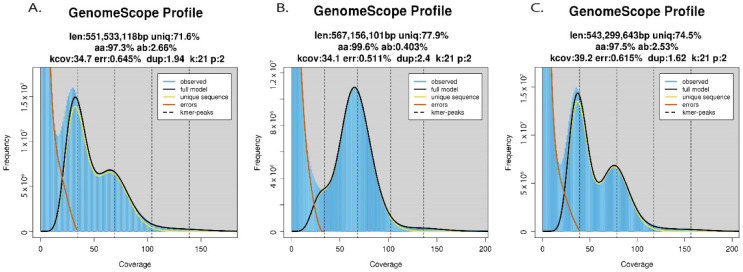
GenomeScope2.0 output showing variation in genome-wide heterozygosity levels among Mexican vanilla accessions: (**A**) MEX41, (**B**) MEX67 and (**C**) MEX79.

**Figure 4 plants-11-02090-f004:**
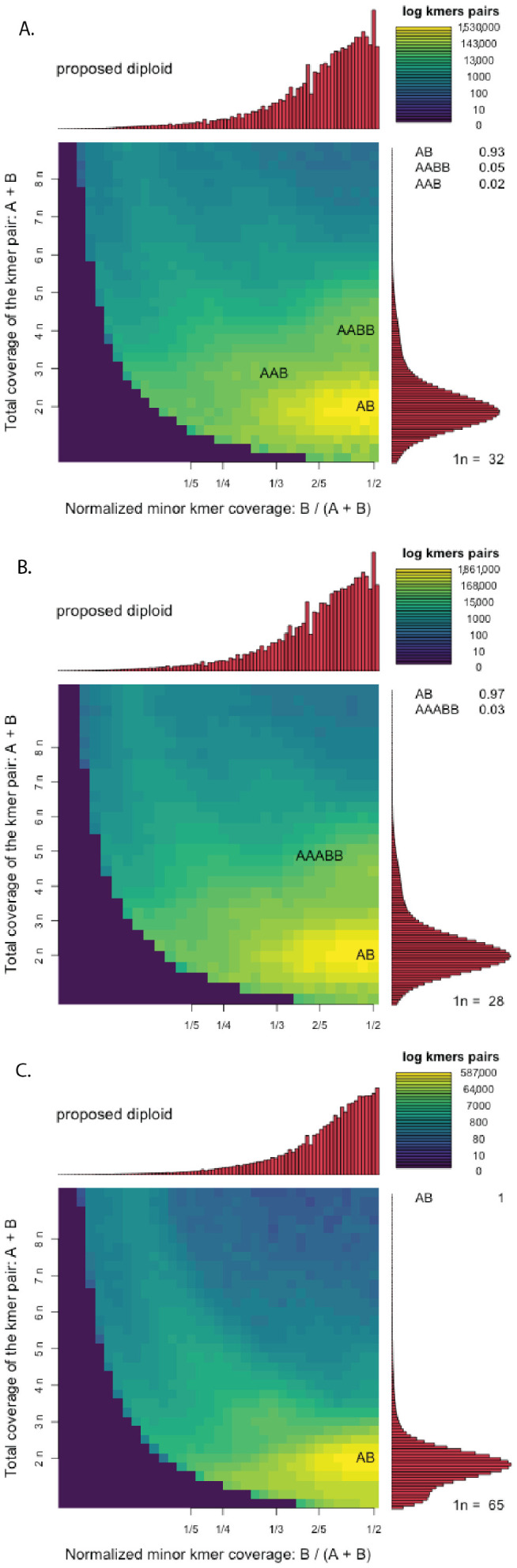
Smudgeplot output for (**A**) MEX13, (**B**) MEX26, and (**C**) MEX67. The color intensity of the smudge indicates how frequently the haplotype structure is represented within each genome and the bar plots represent sequencing coverage.

**Figure 5 plants-11-02090-f005:**
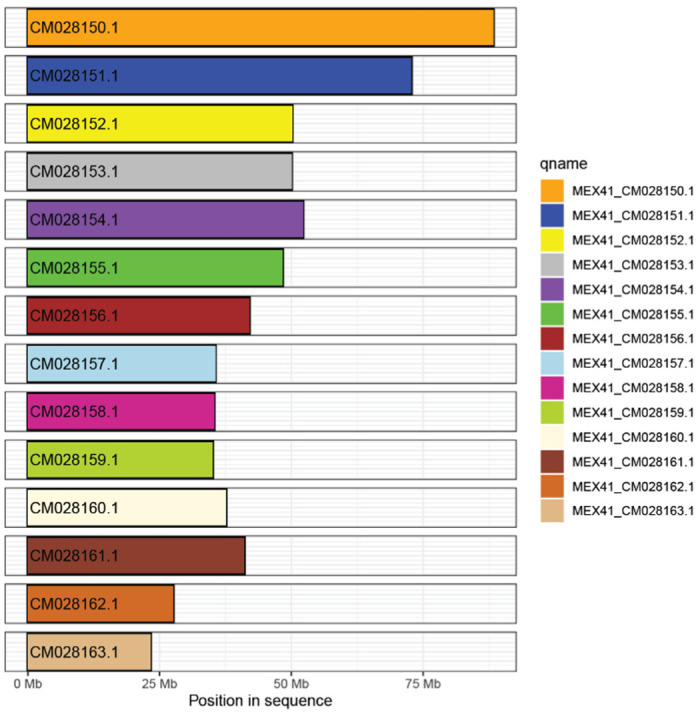
Genome coverage results of MEX41 on the reference genome. Fully colored chromosomes show complete coverage.

**Figure 6 plants-11-02090-f006:**
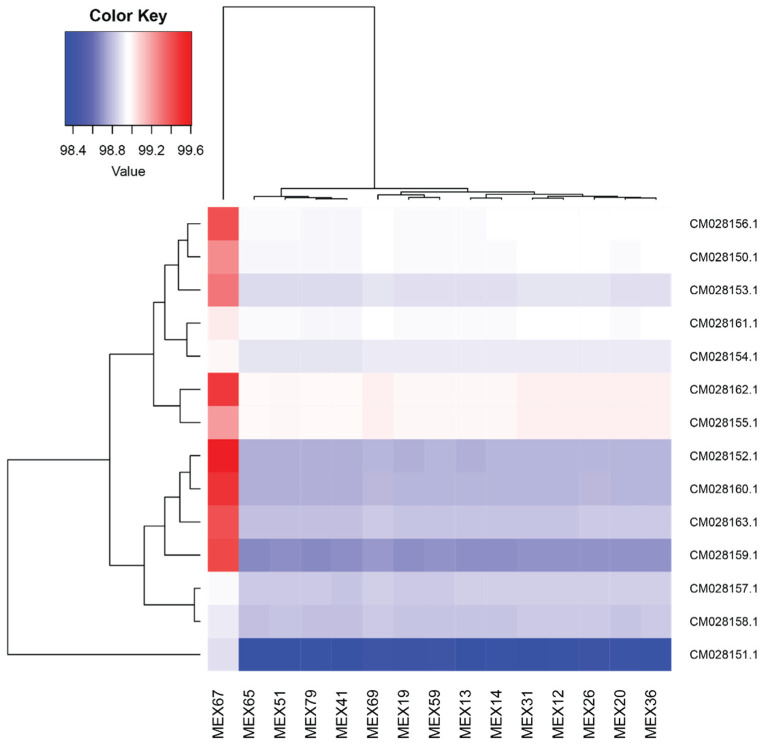
Heat map of chromosomal similarities between Mexican *V. planifolia* samples and reference ‘Daphna’ genome.

**Figure 7 plants-11-02090-f007:**
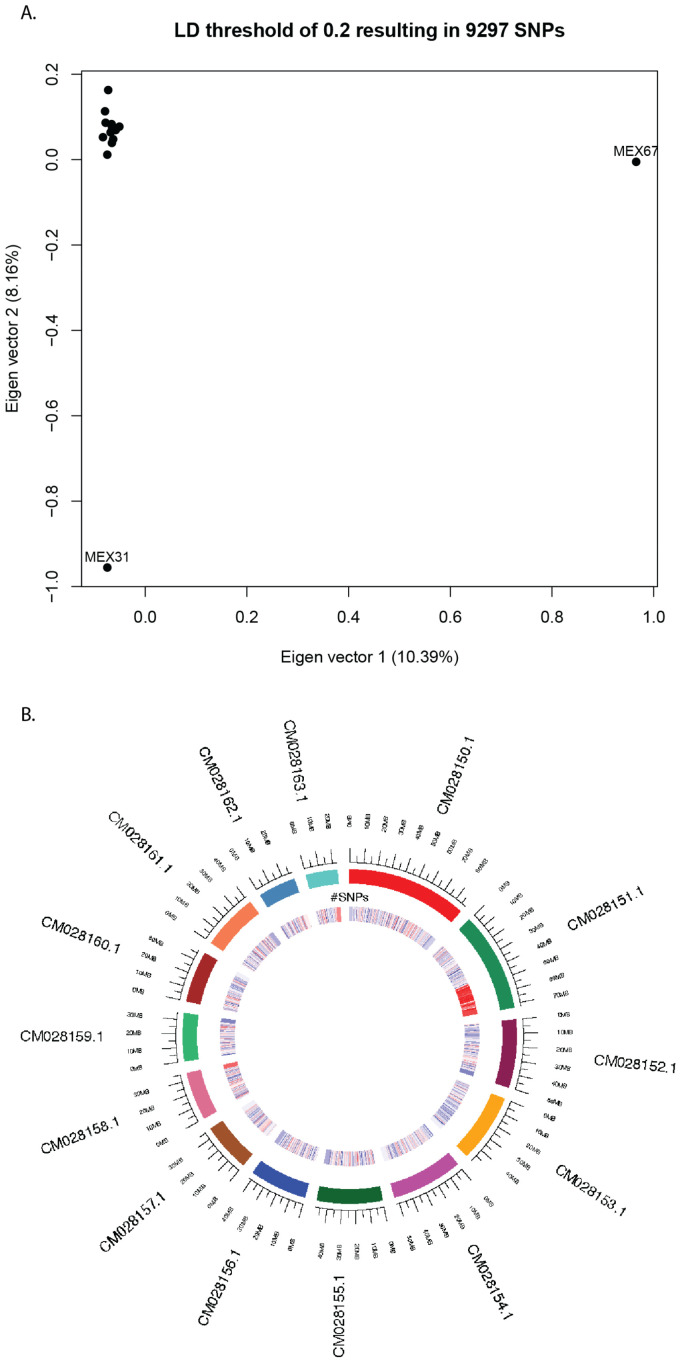
SNP analyses using linkage disequilibrium thresholds set to 0.2 resulting in 9297 SNP markers: (**A**) Principal component analysis (PCA) and (**B**) SNP density along 500 Kb sliding window on 14 chromosomes (colored). Regions of SNP density are illustrated on a color gradient from blue (**low**) to red (**high**).

**Table 1 plants-11-02090-t001:** Attributes and identifiers of generated genomic data used in this study. Raw genome coverage was calculated based on the 1-C reference genome size (736,752,966 bp) from the BioProject PRJNA633886 published in Hasing et al. [20].

Sample_ID	BioSample	SRA	Number of Reads	Yield (Mbases)	Sequencing Coverage (X)
MEX12	SAMN28632720	SRR19374418	187,296,525	5618.896	76.27
MEX13	SAMN28632721	SRR19374411	191,490,389	5744.712	77.97
MEX14	SAMN28632722	SRR19374410	186,065,732	5581.972	75.76
MEX19	SAMN28632723	SRR19374409	169,276,636	5078.299	68.93
MEX20	SAMN28632724	SRR19374408	185,396,535	5561.896	75.49
MEX26	SAMN28632725	SRR19374407	181,142,904	5434.287	73.76
MEX31	SAMN28632726	SRR19374406	187,560,091	5626.803	76.37
MEX36	SAMN28632727	SRR19374405	169,478,017	5084.341	69.01
MEX41	SAMN28632728	SRR19374404	220,584,559	6617.537	89.82
MEX51	SAMN28632729	SRR19374417	215,541,915	6466.257	87.77
MEX59	SAMN28632730	SRR19374416	206,393,194	6191.796	84.04
MEX65	SAMN28632731	SRR19374415	214,461,750	6433.852	87.33
MEX67	SAMN28632732	SRR19374414	194,615,015	5838.450	79.25
MEX69	SAMN28632733	SRR19374413	207,909,587	6237.288	84.66
MEX79	SAMN28632734	SRR19374412	213,402,841	6402.085	86.90

**Table 2 plants-11-02090-t002:** Genomic structure attributes of reference ‘Daphna’ genome and Mexican *V. planifolia* samples, inferred through GenomeScope2.0 and Smudgeplot (ploidy).

Sample ID	Ploidy	Genome-Wide HetErozygosity (ab%)	K-mer Coverage (kcov)
‘Daphna’	2x	2.48	99
MEX12	2x	2.74	32.3
MEX13	2x	2.67	32.6
MEX14	2x	2.66	34.7
MEX19	2x	2.52	32.4
MEX20	2x	2.56	33.7
MEX26	2x	2.57	31.7
MEX31	2x	2.61	32.5
MEX36	2x	2.52	28.2
MEX41	2x	2.77	40.9
MEX51	2x	2.62	39.3
MEX59	2x	2.79	38.5
MEX65	2x	2.57	16
MEX67	2x	0.403	34.1
MEX69	2x	2.85	38.9
MEX79	2x	2.53	39.2

**Table 3 plants-11-02090-t003:** SNP quantity by chromosome based on filtering with linkage disequilibrium (LD) thresholds set to 0.2 and 0.8.

Chromosome Number	Chromosome ID	SNP Quantity (LD Threshold = 0.2)	SNP Quantity (LD Threshold = 0.8)
1	CM028150.1	1248	45,148
2	CM028151.1	1315	129,408
3	CM028152.1	710	27,900
4	CM028153.1	680	23,651
5	CM028154.1	723	24,225
6	CM028155.1	678	23,166
7	CM028156.1	570	19,088
8	CM028157.1	525	19,218
9	CM028158.1	524	17,213
10	CM028159.1	487	23,185
11	CM028160.1	508	21,826
12	CM028161.1	564	17,157
13	CM028162.1	418	14,437
14	CM028163.1	347	14,263
	Total	9297	419,885

## Data Availability

The data produced in this project are deposited on NCBI under BioProject (PRJNA841950) and SRA accessions SRR19374404-SRR19374419 (see Table 1). We are also using data from the BioProject PRJNA633886 published in Hasing et al. [20].

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
