# Peer review of "Genomic Insights into Cultivated Mexican Vanilla planifolia Reveal High Levels of Heterozygosity Stemming from Hybridization"

_plants, 2022, doi:10.3390/plants11162090_

Round 1
Reviewer 1 Report
This work reports the evolutionary mechanisms of the high levels of genome-wide heterozygosity in vanilla and evolutionary history of vanilla in Mexico. I have several concerns which must be taken in account by the authors.
1. The author claimed that the aim of studying the genomic heterozygosity using GenomeScope is obtain a reference scale of genome-wide heterozygosity levels of plants. They studied the genomic heterozygosity of 142 plants which is available on PubMed. This result in not convincing to provide a reference scale for vanilla because there are a lot of open-pollinated plants which have high heterozygosity. Therefore, I suggested that this analysis should use the plants in Orchidaceae family to show the genome-wide heterozygosity.
2. This part of result ‘Sampling, DNA extraction, and whole genome re-sequencing’ has no information of DNA extraction.
3. The author claimed that chromosome-scale genomes could be reconstructed by mapping trimmed reads to the reference genome ‘Daphna’, indicating that there are no chromosomal rearrangements. This conclusion is not convincing. We can say the query and the reference genome do not have presence and absence variations. However, this analysis cannot show the chromosomal rearrangements by only aligning illumina reads to the reference genome.
4. The font size of all figures (Figure 1, Figure 3, Figure 4) is too small. Please increase the font size and make the legend readable.
Author Response
- The author claimed that the aim of studying the genomic heterozygosity using GenomeScope is obtain a reference scale of genome-wide heterozygosity levels of plants. They studied the genomic heterozygosity of 142 plants which is available on PubMed. This result in not convincing to provide a reference scale for vanilla because there are a lot of open-pollinated plants which have high heterozygosity. Therefore, I suggested that this analysis should use the plants in Orchidaceae family to show the genome-wide heterozygosity.
Authors’ Response:
- Of all studies which have used GenomeScope to infer genomic heterozygosity, only four reported on orchids (including planifolia). Therefore, it was not possible to create a substantial reference for this family. We mention the fact that many of these reported species may exhibit highly outbreeding reproductive systems, however, that the scale is likely biased to the lower extreme due to the fact that many crop genomes are assessed from highly inbred accessions which facilitates genomic analyses and constructions. To highlight the levels of other orchids, however, we have added their labelling to the violin plot (Figure 1A).
- Revised Lines 132-134: “Over half of the plants assessed in these studies were cultivated for human use (Figure 1B) and Orchidaceae was only represented by three other species (Figure 1A)”
- This part of result ‘Sampling, DNA extraction, and whole genome re-sequencing’ has no information of DNA extraction.
Authors’ Response:
- We have revised the Methods to include the genomic DNA quality threshold (as inferred with the Qubit fluorescence protocol to check for sufficient concentration prior to sequencing) and revised the Results to refer to the Methods.
- Revised Line 355: “Extracted genomic DNA with concentrations greater than 20 ng/µL was sent to GENEWIZ, inc.”
- Revised Lines 152-153: “Whole genome re-sequencing of quality extracted genomic DNA (see Methods for DNA concentration threshold) resulted in an …”
- The author claimed that chromosome-scale genomes could be reconstructed by mapping trimmed reads to the reference genome ‘Daphna’, indicating that there are no chromosomal rearrangements. This conclusion is not convincing. We can say the query and the reference genome do not have presence and absence variations. However, this analysis cannot show the chromosomal rearrangements by only aligning illumina reads to the reference genome.
Authors’ Response:
- We agree that this sentence was not clear and have revised the sentence.
- Revised Lines 192-195: “Genomic alignments from MiniMap2 revealed that all reconstructed genomes exhibited full coverage on the ‘Daphna’ reference genome (Figure 5) and, in congruence with results from the dotplot analyses (Figure S3), suggested no chromosomal rearrangements among the accessions.”
- The font size of all figures (Figure 1, Figure 3, Figure 4) is too small. Please increase the font size and make the legend readable.
Authors’ Response:
- We have increased the font size of Figures 1 and 3 and reformatted Figure 4 so that the font is larger.
Reviewer 2 Report
The presented article is devoted to the analysis of heterozygosity of vanilla genotypes. Based on the analysis of NGS data, the researchwork shows both differences in the level of heterozygosity and differences in the SNP pattern compared to the reference genome.These results do not raise questions for me.
However, the reconstruction of haplotypes is not obvious to me from the text of the article. Why haplotypes A and B have been identified for different genotypes? In phylogenetic analysis, we see several clusters. An alternative explanation would be a different combination of haplotype A with B and C or even D. These results and their discussion require more clarity.
Author Response
The presented article is devoted to the analysis of heterozygosity of vanilla genotypes. Based on the analysis of NGS data, the researchwork shows both differences in the level of heterozygosity and differences in the SNP pattern compared to the reference genome.These results do not raise questions for me.
However, the reconstruction of haplotypes is not obvious to me from the text of the article. Why haplotypes A and B have been identified for different genotypes? In phylogenetic analysis, we see several clusters. An alternative explanation would be a different combination of haplotype A with B and C or even D. These results and their discussion require more clarity.
Authors’ Response:
Results from Smudgeplot only allow us to discuss the two sub-genomes that make up this diploid organism and not the presence of additional haplotypes as was inferred through other analyses. We included the term “sub-genome” and additional descriptions of the Smudgeplot results within the discussion to make this distinction clearer.
- Revised Lines 255-258: “For V. planifolia, however, results from Smudgeplot confirmed the diploid status of all accessions by uncovering almost exclusive AB k-mer pairs (average 97.3%; Figure 4), indicating the prominence of two sub-genomes, and therefore ruling out these latter evolutionary events as sources of the observed high heterozygosity”
- Revised Lines 281-283: “These results indicate that multiple haplotypes exist within the AB sub-genomes identified through Smudgeplot analyses (Figure 4)”
Round 2
Reviewer 1 Report
Thanks for your response. I do not have any further questions.